# Effect of Targeted Embolization on Seizure Outcomes in Patients with Brain Arteriovenous Malformations

**DOI:** 10.3390/diagnostics13010047

**Published:** 2022-12-23

**Authors:** Chingiz Nurimanov, Aisha Babi, Karashash Menlibayeva, Yerbol Makhambetov, Assylbek Kaliyev, Elena Zholdybayeva, Serik Akshulakov

**Affiliations:** 1Vascular and Functional Neurosurgery Department, National Center for Neurosurgery, Astana 010000, Kazakhstan; 2Hospital Management Department, National Center for Neurosurgery, Astana 010000, Kazakhstan; 3National Center for Biotechnology, Astana 010000, Kazakhstan

**Keywords:** brain arteriovenous malformation, embolization, seizure, ILAE, Engel

## Abstract

Background: Seizures are one of the most debilitating manifestations of brain arteriovenous malformations (AVMs). This study aimed to evaluate the effect of curative embolization on brain AVM patients presenting with seizures. Methods: The records of patients who underwent embolization for brain AVM from January 2012 to December 2020 were evaluated and patients presenting with seizures were interviewed. Patient responses were evaluated according to the International League Against Epilepsy (ILAE) and Engel classifications. Statistical analyses of factors associated with seizure outcomes and complications were performed using ANOVA and Fischer’s exact tests. Results: The mean age of the participants was 35.2 ± 10.7 years. More than 80% of the patients received no or suboptimal dosages of antiepileptic drugs (AEDs) prior to embolization. Positive seizure dynamics were observed in 50% of the patients post-procedure. A correlation was found between length of seizures in anamnesis and outcomes of both Engel and ILAE score, where shorter length was associated with better outcomes. Post-embolization hemorrhage was associated with initial presentation with hemorrhage. Conclusions: The embolization of brain AVMs had a positive effect on seizure presentation and a relatively low prevalence of complications. However, the results of the study are obscured by inadequate AED treatment received by the patients, which prompts prospective studies on the topic with careful patient selection.

## 1. Introduction

Intracranial arteriovenous malformations (AVMs) are the abnormal development of the arteries and veins of the brain [1]. Cerebral AVMs can present with hemorrhage, seizures, and neurological deficits, but can also be discovered incidentally and present with no symptoms. A life-threatening hemorrhage occurs when an AVM ruptures, which has morbidity and mortality rates of 45% and 20%, respectively [2,3]. Seizure is the second most frequent initial sign of brain AVMs after hemorrhage and it occurs in 33 to 58% of all AVM cases [4]. Seizures can be focal, generalized, and unclassified and may indicate a lesion in the brain [5].

AVMs presenting with seizures are a huge neurological disorder that considerably affects patients’ neuropsychological status and quality of life and require proper and timely treatment. Larger AVM size [6] and high Spetzler–Martin (SM) grade [7] have been found to be associated with epilepsy. There are interventional and conservative managements of epileptogenic brain AVMs. Interventional treatment includes microsurgical resection, stereotactic radiosurgery, endovascular embolization, or a combination of the above [8]. The choice of AVM treatment depends on AVM characteristics, such as location and grade, patient characteristics, and surgeon skills.

The microsurgical intervention has significantly more positive outcomes for small, unruptured, low-SM-grade AVM when compared to multimodal treatment or conservative treatment [9]. In addition, among all the interventional treatment options microsurgery showed the best results for patients with AVM-associated seizures [10,11]. Noninterventional medical treatment is considered for unruptured incidentally discovered brain AVMs. In the cases of brain AMVs presented with seizures, interventions have shown better outcomes with post-operative seizure freedom of 50–78% [12]. Stereotactic radiosurgery has much lower rates of complications and mortality when compared to other treatment modalities [13]. Despite low rates of complication, stereotactic radiosurgery has varying results in clinical outcomes and is the most optimal for small AVMs located in difficult-to-reach areas of the brain [14]. In the treatment of AVM-associated seizures, stereotactic radiosurgery has been most effective in seizure control when the AVM was successfully obliterated [15].

However, in cases where the above-mentioned methods are not applicable due to the anatomy of the AVM or due to the unavailability of the equipment in the case of SRS, the only interventional method remaining is endovascular embolization. The effect of embolization alone on AVM patients presenting with seizures is largely understudied; as a recent, meta-analysis included only three studies that used endovascular methods to cure AVM-related seizures [16]. A few reports evaluating the seizure outcomes have different patient selection criteria and results ranging from 50% [17] to 100% [18]. As was concluded by Mosimann et al. [19], the effectiveness of embolization alone on the curation of brain AVM is uncertain and requires more studies. Therefore, this study aimed to retrospectively evaluate the use of embolization alone on the seizure outcomes in AVM patients based on the data from the largest neurosurgical hospital in the country to provide more insight on endovascular embolization as a treatment option.

## 2. Materials and Methods

### 2.1. Study Design and Participants

This is a retrospective cross-sectional study of patients who underwent one or more sessions of endovascular embolization during the period of 8 years from January 2012 to December 2020 at the National Center for Neurosurgery (NCN, Astana, Kazakhstan). The interventional treatment of AVMs was the standard approach before the final report of the ARUBA study had been published. Therefore, whenever embolization could be safely performed on AVM patients it was chosen over medical management alone.

A total of 421 patients who underwent embolization were reviewed for eligibility (Figure 1). Among those patients, 66% had relevant contact information and were reached out to with a questionnaire. A total of 97 patients were excluded due to incomplete follow-up, and 179 patients provided full information. Fifty-four patients did not experience seizures before the surgery and were removed from the study. Nine patients have passed away; however, the cause of death for seven of the patients is unknown. A total of 116 AVM patients who have suffered from seizures were included in the analysis. Such characteristics as age, sex, AVM location, type of seizure, years with seizures, antiepileptic drug administration, dosage, SM grade, type (partial or total) of embolization, clinical presentation, and operative and post-operative complications were collected.

### 2.2. Diagnosis of Brain AVMs

Brain AVMs were diagnosed by magnetic resonance imaging (MRI) and digital subtraction angiography (DSA) using biplane (Artis Zee Biplane System; Siemens, Erlangen, Germany). Images were examined by a multidisciplinary team of neuroradiologists and neurosurgeons.

### 2.3. Endovascular Embolization Procedure

The main criteria for embolization were the following angiographic findings: a direct AV fistula within the AVM, dilated varix, and afferent artery with a diameter 2–3 times wider than the arteries supplying this region of the brain (Figure 2). The brain AVMs not eligible for neurosurgical intervention according to the results of angiography were kept under clinical observation.

Endovascular embolization was performed under general anesthesia on a biplane angiographic unit via the transarterial route. Vascular access was achieved via the femoral artery using a 6F guiding catheter. We used microcatheters with detachable tips: Apollo or Sonic and embolic agents such as Onyx, PHIL, or Squid. After partial embolization or complete embolization of the AVM nidus, the microcatheter was removed. The embolization procedures were terminated by post-embolization angiography via the guide catheter. The main goal of the embolization procedures was to reduce the risk of hemorrhage via partial embolization and total embolization where possible.

### 2.4. Inclusion and Exclusion Criteria

Patients who were without initial seizures had AVMs located on the cerebellum, were treated with methods other than endovascular embolization, and had a follow-up period of fewer than 9 months were excluded from the study. Incomplete follow-up responses with missing information were not considered for the analysis.

### 2.5. Follow-Up

Follow-up was performed via a questionnaire consisting of 12 questions administered through a telephone interview. Information collected included the status of seizures, their characteristics (generalized or focal), dynamic frequency, presence of aura, medication’s name, and dose if available. Patients were contacted via phone and were interviewed by neurologists. Follow-up data were analyzed by a neurologist and classified according to the International League Against Epilepsy (ILAE) classifications of epilepsy surgery seizure outcome and Engel classification of seizure outcomes. In this study, ILAE classes 1 and 2 were grouped into a positive outcome, classes 3 and 4—moderate outcome, and classes 5, and 6—negative outcome. The Engel classes were grouped according to classes: class 1—free of disabling seizures, class 2—almost seizure-free, class 3—worthwhile improvement, and class 4—no worthwhile improvement.

### 2.6. Ethical Consideration

The study was approved by the Institutional Review Board of the National Center for Neurosurgery, protocol #5, dated 25 November 2021. Written informed consent forms were obtained from all participants.

### 2.7. Statistical Analysis

All the data were cleaned and coded using Excel (Microsoft Office) and further analyzed with STATA software (Version 16.0; Stata Corporation, College Station, TX, USA). Data analysis consisted of descriptive statistics, such as percentages, frequencies, and means with standard deviation. Clinical outcomes were tested for association with independent variables with one-way analysis of variance (ANOVA) and Fisher’s exact test, where appropriate. A *p*-value of less than 0.05 was considered statistically significant.

## 3. Results

A total of 116 patients have been included in the study. The average follow-up was 53.6 ± 23.7 months. The mean age of the participants was 35.2 ± 10.7 years, and about 50% of the sample was younger than 33 y.o. Most of the sample were males (56.9%). The majority of the sample had suffered from seizures for 10 years or less, and approximately 24% had seizures for more than 10 years. The most common AVM locations were the temporal region (25%) and the left hemisphere (17.2%). The majority of the patients had SM grading 3 (45.7%), followed by grade 2 (27.6%), grade 4 (17.2%), grade 1 (5.2%), and grade 5 (4.3%). The patients mainly had a generalized onset of seizures (47.4%), and approximately a third had focal onset (33.6%). Only 15.5% of the patients had achieved complete embolization of the 116 AVMs; the rest of the patients had only partial embolization

After the embolization, 50% of the sample achieved Engel class 1, and 31% achieved Engel class 2. A total of 6% had class 3, and 13% did not see worthwhile improvement, ending up with class 4. According to the ILAE score, 52% of the sample had a positive result, meaning the complete absence of seizures and aura or aura only. Moderate outcomes were achieved by 40% of the sample, where some, but not complete improvement with seizures was observed. Almost 9% of the sample experienced negative results, where the improvement of the seizures was not significant enough or the seizures became worse (Table 1).

All patients had experienced seizures before the surgery (Table 2). The majority (37%) had less than five seizures per year. However, almost 21% had 11–20 seizures per year, and 21% had more than 20 seizures per year. After the surgery, 47% of the sample got rid of seizures completely, 3% had between 11 and 20 seizures per year, and 6% had more than 20. The number of patients who experienced auras (35%) had also decreased to approximately 20%. Before the surgery, only 63% of the sample had regularly taken prescribed antiepileptic medication and the number changed by 1% after the surgery. Both before and after surgery, the most popular medication prescribed was carbamazepine, followed by sodium valproate. Before the embolization, only 2% of the sample were on two or more medications, but after the surgery, the proportion increased to 4%. Before the embolization, 11% of the sample were taking optimal/maintenance doses of the AED. The majority (47%) were medicated below the optimal dose, and none would qualify to have an AED resistance. After the surgery, the proportion of patients with optimal doses of AEDs increased to 23%.

During the follow-up, it was discovered that nine patients have passed away. Two of those patients have died because of the surgery. The cause of the death of the other patients is unknown. Therefore, the mortality rate is anywhere between 2% and 7%. The deceased patients were not included in the analysis. After the surgery, bleeding was experienced by 15 (13%) patients. Bleeding after the embolization procedure was associated with a history of previous hemorrhage (*p*-value = 0.013) but was not associated with other clinical and demographic characteristics (Table 3). Eight patients had complications during the surgery, which resulted in hemiparesis on the left side, lower right monoparesis, hypertensive syndrome, extravasation, homonymous hemianopia, and stenosis of the draining vein. No correlation was found between patient characteristics and the outcomes of both Engel and ILAE scores, except for the length of suffering from seizures. The longer the patient has experienced seizures, the worse the outcomes from embolization were (Table 4).

## 4. Discussion

Brain AVM embolization is mostly used in combination with other treatment options, such as microsurgery and SRS. Curative embolization is a topic that has been studied sporadically with differing results. Obliteration rates reported so far have varied from 4% in earlier reports up to 100%, which mainly depends on the selection of the patients, the embolization technique, the anatomy of the AVM, and the intent to cure, as was summarized by Ghali et al. The complication rates also vary significantly, as was summarized previously, where mortality ranges from 0 to 4.3% and morbidity from 0 to 22% [21].

The effect of embolization on AVM-caused seizures is reported more rarely. A 2010 retrospective study based on 30 unruptured brain AVM patients who were resistant or unable to take AED and were treated with embolization demonstrated excellent results in 70% of the sample. The follow-up period ranged from 2 years up to 8 years. Total obliteration was achieved in 13% of the patients. Ten percent of the patients have experienced bleeding during or after the surgery, and with other neurological deficits, the total rate of complications was 23% [18]. A more recent study of 37 patients presenting with seizures and treated with embolization alone or embolization followed by SRS reported 51.4% of participants to be seizure-free 12 months after the procedure [22]. The rate of bleeding during or after the surgery was 5% and the overall rate of complications was 11%.

In our study, according to both ILAE and Engel classification, around 50% of the patients were seizure-free after embolization, which is similar to previous literature, despite embolization alone being used as an intervention among our patients. Total obliteration was achieved in 16% of the patients, which is similar to the results of Lv et al. [18]. The rate of post-operative bleeding was 13%, and 7% developed neurological deficits. During the follow-up, two patients were established to be deceased from the surgery and seven more died of unspecified causes, making the mortality rate between 2 and 6%.

Among our patients, there was an association of embolization outcomes with the years that the patient suffered from seizures. The longer the patient had suffered from seizures, the worse the outcomes of the surgery. It is unclear why the duration of epilepsy influences the surgical result, but some authors have reported that patients with chronic epilepsy are more susceptible to developing neocortical atrophy [23]. Other clinical studies supposed that patients who suffered from epilepsy for a long time have structural and functional changes in other brain areas which do not include the primary epileptogenic zone, and this could be associated with the persistence of disabling seizures [24,25]. Similar findings regarding years with seizures were observed by Yeh et al. [10], but the patients were treated with microsurgical intervention.

Among the studied patients, the only factor that showed significant association with the risk of post-procedural hemorrhage was previous presentation with hemorrhage, which is a known risk factor [26]. Other factors, such as age, sex, location, and AVM grade did not differ between those who experienced bleeding and those who did not. Meanwhile, in a study evaluating the risk of rupture after the partial embolization of brain AVMs [27], no association of hemorrhage during follow-up with an initial presentation with hemorrhage was found. However, the study has found patients with deep venous drainage and an SM grade of 3–5 to have a higher proportion of hemorrhage after the embolization. The mismatch in the results could be attributed to the different approaches in AVM embolization, where in the study by Lv et al. a fifth of patients were further treated with SRS, were embolized only partially, and had a higher proportion of SM grade 3–5 patients.

In our study, 37% of the patients did not take AED before the embolization procedure, which is similar to the findings by Zhang et al. [22] who reported that among 37 patients presenting with epilepsy 38% did not take AED. In their study, the number after the embolization procedure increased to 43%. However, in the current study, the percentage of patients taking AED has decreased by 1% only, despite 50% of the patients becoming seizure-free.

This could be explained by the initial suboptimal dosage of AED among those who took the medications. Only 11% of the sample had taken the optimal dosage of the corresponding AED before the surgery. After the surgery, the proportion increased to 23%. None of the patients have taken a dosage above the maintenance dose of the drug and could not be classified as drug-resistant epilepsy patients. Moreover, only 2% of the patients pre-surgery and 4% after surgery were administered a combination of two or more AEDs. Our results correspond with the findings of a study based on data from pharmacy centers in Kazakhstan (N = 57,959), where only 8.3% of the patients received two or more AEDs at the same time [28]. Since 30–40% of the patients do not reach control of seizures with monotherapy alone [29], the lack of polytherapy points to poor adherence, prescription, or availability of AEDs in the country.

Our findings align with the findings of the study from the southern part of Kazakhstan, where a significant difference in medication uptake was found between urban and rural populations. A significant portion of rural epilepsy patients (30%) did not take any AEDs. In both urban and rural populations, around 16% of the patients were on polytherapy. Moreover, the dosage of the treatment received was often suboptimal [30], similar to our findings, where 47% of the patients before surgery took a suboptimal dosage and 40% after the surgery. The findings agree with the situation in another post-Soviet country, Georgia. The researchers have found that only 34% of the participants with epilepsy were on appropriate AED treatment [31]. Since NCN admits patients from all over the country through Compulsory Social Health Insurance, a large proportion of the patients come from rural areas. This coincides with the findings in Georgia, where around 50% of epilepsy patients are socially disadvantaged, which also might be the case with rural patients in Kazakhstan. The true effect of embolization on AVM-related seizures in our study is unclear, due to inadequate adherence to or prescription of AEDs among the patients.

The exact mechanisms of seizures by AVMs remain unclear, but some articles have suggested that the main risk factors associated with epilepsy are male sex, age younger than 65, AVM size of over 3 cm, and location in the temporal lobe [32]. Shankar and colleagues have also found an association between AVM size and seizure presentation [33]. This could result from the AVM-related hypoxic condition in brain tissue, which leads to seizures [33], or due to the mass effect, which results in large venous ectasia or the nidus proper compressing critical structures [34]. Larger AVMs have more arteriovenous shunting of blood, vascular steal-induced epilepsy associated with focal cerebral ischemia [33]. We reckon that the improvement of seizure presentation in our patients is due to the decrease in vascular steal syndrome, and the reduced mass effect of dilated AVM vessels after the embolization procedure.

Despite the conclusions of the ARUBA study that suggest medical management is superior to surgical intervention in unruptured AVMs [35], our study has shown only 11% of the primary ARUBA outcomes (hemorrhage and death) in patients with unruptured AVMs. Our results align with studies performed post-ARUBA that indicate better outcomes from intervention when compared to the ARUBA medical management group [36], and studies that report a decrease in endovascular procedures and an increase in ruptured AVMs post-ARUBA [37]. Overall, the results of our study with the backdrop of the recent findings indicate that, with careful selection of the patients, endovascular treatment for AVM-related seizure is a valid option. However, the propensity for seizure patients in Kazakhstan to remain under- or untreated medicinally and opt for surgery, when not all pharmacological treatment options were attempted, cannot be ignored.

There are several limitations to this study. First, the retrospective nature of the study limits the information available for the analysis. Second, the outcomes of the procedure were self-reported, which could create self-selection and survivorship bias. Although the response rate in the study was 66%, the rate of complete follow-up was only 43%. Moreover, the follow-up period was unstructured and ranged from 10 to 97 months with a mean of 53.6 ± 23.7 months. Finally, the selection of patients was nonrandom based on the compatibility of the procedure with risk–benefits outcomes for the patients. There is a need for a prospective randomized study that accesses the embolization of AVM with seizures as one of the morbidity factors.

## 5. Conclusions

The embolization of brain AVM patients presenting with seizures resulted in seizure freedom or seizure improvement in most patients, and under careful consideration of the patient’s characteristics could be used as a treatment option. The outcomes of embolization were associated with the length of seizures in anamnesis with shorter length resulting in better outcomes. Post-procedural bleeding was associated with initial hemorrhage presentation. Lack of adherence to adequate AED treatment impedes the conclusion on the effectiveness of embolization as an AVM-related seizure treatment option. Further prospective studies on AVM embolization, as well as on the epilepsy treatment gap in Kazakhstan, are required.

## Figures and Tables

**Figure 1 diagnostics-13-00047-f001:**
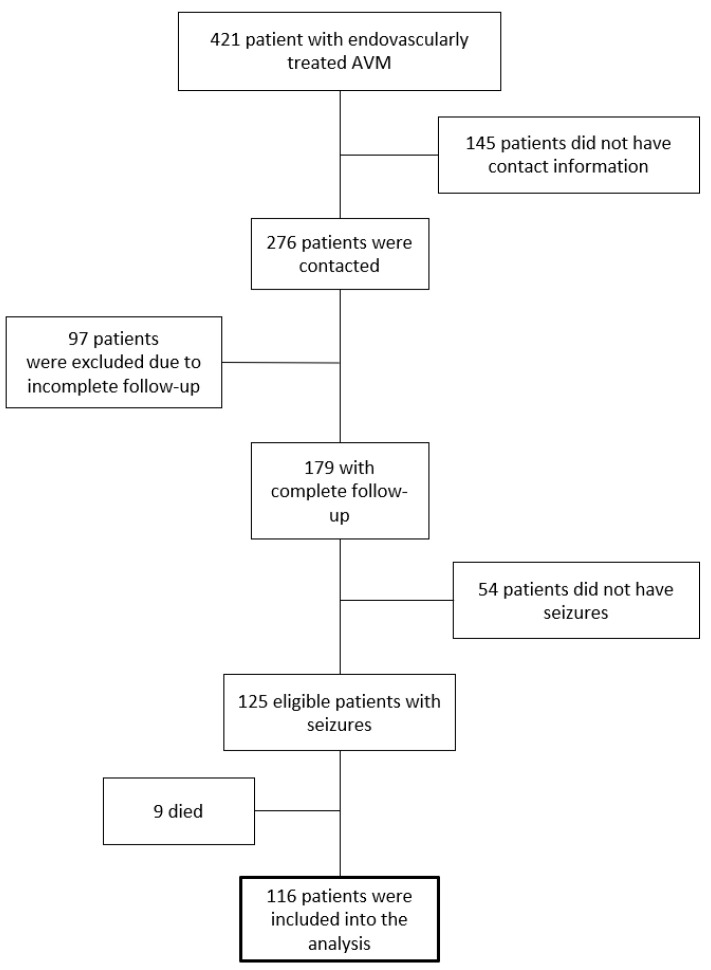
Patient exclusion and inclusion flow chart.

**Figure 2 diagnostics-13-00047-f002:**
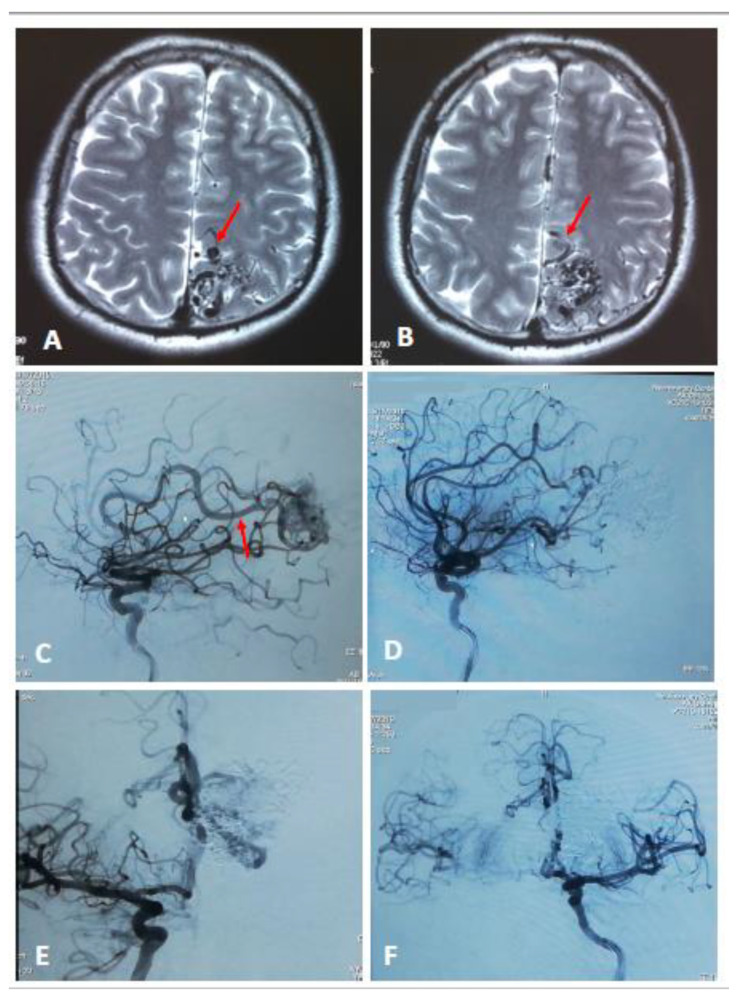
Axial MRI of grade 3 brain AVM demonstrates varix (**A,** red arrow) and dilated afferent vessel (**B,** red arrow). Lateral view angiogram demonstrates dilated afferent artery before embolization (**C,** red arrow) and after (**D**). Anterior-posterior view angiogram before total embolization (**E**) and after (**F**).

**Table 1 diagnostics-13-00047-t001:** Demographic and clinical characteristics of participants.

Variable	N (%)
Age	
Mean ± SD	35.2 ± 10.66
<29	37 (31.9)
29–33	24 (20.69)
34–41	28 (24.14)
>41	27 (23.28)
Sex	
Male	66 (56.9)
Female	50 (43.1)
Spetzler–Martin Grade	
1	6 (5.17)
2	32 (27.59)
3	53 (45.69)
4	20 (17.24)
5	5 (4.31)
Location	
Temporal	29 (25)
Hemisphere left	20 (17.24)
Hemisphere right	15 (12.93)
Parietal	18 (15.52)
Occipital	13 (11.21)
Deep structure	5 (4.31)
Frontal	16 (13.79)
Engel class	
1	58 (50)
2	36 (31.03)
3	7 (6.03)
4	15 (12.93)
ILAE score	
Positive	60 (51.72)
Moderate	46 (39.66)
Negative	10 (8.62)
Seizure type	
Focal onset	39 (33.62)
Generalized onset	55 (47.41)
Unclassified	22 (18.97)
Embolization	
Partial	98 (84.48)
Total	18 (15.52)
Bleeding after surgery	
No	101 (87.07)
Yes	15 (12.93)
Clinical manifestation	
Seizure only	81 (69.83)
Combination with hemorrhage	35 (30.17)
Years with seizures	
<5	29 (25.22)
5–10	59 (51.3)
>10	27 (23.48)

**Table 2 diagnostics-13-00047-t002:** Seizure manifestation and management pre- and post-embolization.

Variables	Pre-Embolization	Post-Embolization
Number of seizures per year		
0	-	55 (47.41)
<5	43 (37.07)	40 (34.48)
5–10	25 (21.55)	11 (9.48)
11–20	24 (20.69)	3 (2.59)
>20	24 (20.69)	7 (6.03)
Presence of aura		
Yes	41 (35.34)	23 (19.83)
No	75 (64.56)	93 (80.17)
Treatment with AED		
Yes	73 (62.93)	74 (63.79)
No	43 (37.07)	42 (36.21)
Drug type		
Carbamazepine	54 (46.55)	57 (49.14)
Sodium valproate	9 (7.76)	4 (3.45)
Levetiracetam	1 (0.86)	3 (2.59)
Lamotrigine	2 (1.72)	3 (2.59)
Oxcarbazepine	-	1 (0.86)
Two or more drugs	2 (1.72)	5 (4.31)
None	43 (37.07)	42 (36.21)
AED Dosage *		
Low	54 (46.55)	46 (39.66)
Optimal	13 (11.21)	27 (23.28)
High	-	-
Does not remember	6 (5.17)	1 (0.86)
None	43 (37.07)	42 (36.21)

*—dosage was categorized according to the maintenance dosage indicated in the clinical protocol for epilepsy treatment curated by the Republican Center for Health Development of the Ministry of Health of the Republic of Kazakhstan [20], where dosage below maintenance was considered low, and dosage above maintenance was considered high (maintenance doses: carbamazepine 600–1200 mg/day, sodium valproate 1000–3000 mg/day, valproic acid 1000–3000 mg/day, levetiracetam 1000–3000 mg/day, lamotrigine 100–200 mg/day, oxcarbazepine 900–2400 mg/day, topiramate 200–400 mg/day).

**Table 3 diagnostics-13-00047-t003:** Factors associated with bleeding after embolization.

	Bleeding after Embolization		
Variable	Yes, N = 15 (12.93%)	No, N = 101 (87.07%)	*p*-Value
Spetzler–Martin Grade			
1	2 (33.33)	4 (66.67)	0.11
2	2 (6.25)	30 (93.75)	
3	7 (13.21)	46 (86.79)	
4	2 (10)	18 (90)	
5	2 (40)	3 (60)	
Clinical presentation			
Hemorrhage and seizures	9 (25.71)	26 (74.29)	0.013
Seizures only	6 (7.41)	75 (92.59)	
Age			
<29	6 (16.22)	31 (83.78)	0.87
29–33	2 (8.33)	22 (91.67)	
34–41	4 (14.29)	24 (85.71)	
>41	3 (11.11)	24 (88.89)	
Sex			
Female	8 (16)	42 (84)	0.42
Male	7 (10.61)	59 (89.39)	
Embolization			
Partial	14 (14.29)	84 (85.71)	0.46
Total	1 (5.56)	17 (94.44)	
Seizure type			
Focal	4 (10.26)	35 (89.74)	0.33
General	6 (10.91)	49 (89.09)	
Unknown	5 (22.73)	17 (77.27)	
Location			
Temporal	3 (10.34)	26 (89.66)	0.55
Hemisphere left	4 (20)	16 (80)	
Hemisphere right	2 (13.33)	13 (86.67)	
Parietal	3 (16.67)	15 (83.33)	
Occipital	2 (15.38)	11 (84.62)	
Deep structure	1 (20)	4 (80)	
Frontal	0	16 (100)	
Years with seizures			
<5	4 (13.79)	25 (86.21)	0.21
5–10	5 (8.47)	54 (91.53)	
>10	6 (22.22)	21 (77.78)	

**Table 4 diagnostics-13-00047-t004:** Demographic and clinical factors associated with seizure outcomes after AVM embolization.

	ILAE Score			Engel Class			
Variable	Positive N = 60 (51.72%)	Moderate N = 46 (39.66%)	Negative N = 10 (8.62%)	1 N = 58 (50%)	2 N = 36 (31.03%)	3 N = 7 (6.03%)	4 N = 15 (12.93%)
Age							
Mean ± SD	34.28 ± 9.92	35.74 ± 11.63	38.1 ± 10.71	34.62 ± 10.07	35.36 ± 11.71	36 ± 12.61	36.6 ± 10.29
*p*-value			0.53				0.93
<29	19 (51.35)	16 (43.24)	2 (5.41)	18 (48.65)	13 (35.14)	2 (5.41)	4 (10.81)
29–33	13 (54.17)	9 (37.5)	2 (8.33)	11 (45.83)	7 (29.17)	3 (12.5)	3 (12.5)
34–41	18 (64.29)	8 (28.57)	2 (7.14)	19 (67.86)	6 (21.43)	0	3 (10.71)
>41	10 (37.04)	13 (48.15)	4 (14.81)	10 (37.04)	10 (37.04)	2 (7.41)	5 (18.52)
*p*-value	0.52			0.48			
Sex							
Female	25 (50)	22 (44)	3 (6)	22 (44)	18 (36)	5 (10)	5 (10)
Male	35 (53.03)	24 (36.36)	7 (10.61)	36 (54.55)	18 (27.27)	2 (3.03)	10 (15.15)
*p*-value			0.61				0.25
Location							
Parietal	12 (66.67)	5 (27.78)	1 (5.56)	11 (61.11)	6 (33.33)	0	1 (5.56)
Deep structures	2 (40)	2 (40)	1 (20)	2 (40)	1 (20)	0	2 (40)
Frontal lobe	6 (37.5)	10 (62.5)	0	6 (37.5)	7 (43.75)	1 (6.25)	2 (12.5)
Left hemisphere	9 (45)	8 (40)	3 (15)	9 (45)	6 (30)	2 (10)	3 (15)
Occipital lobe	7 (53.85)	3 (23.08)	3 (23.08)	6 (46.15)	4 (30.77)	0	3 (23.08)
Right hemisphere	6 (40)	8 (53.33)	1 (6.67)	7 (46.67)	4 (26.67)	1 (6.67)	3 (20)
Temporal lobe	18 (62.07)	10 (34.48)	1 (3.45)	17 (58.62)	8 (27.59)	3 (10.34)	1 (3.45)
*p*-value			0.25				0.75
Spetzler–Martin Grade							
1	3 (50)	0	3 (50)	3 (50)	2 (33.33)	0	1 (16.67)
2	18 (56.25)	12 (37.5)	2 (6.25)	17 (53.13)	11 (34.38)	1 (3.13)	3 (9.38)
3	29 (54.72)	19 (35.85)	5 (9.43)	29 (54.72)	16 (30.19)	3 (5.66)	5 (9.43)
4	9 (45)	10 (50)	1 (5)	8 (40)	7 (35)	3 (15)	2 (10)
5	1 (20)	2 (40)	2 (40)	1 (20)	0	0	4 (80)
*p*-value			0.52				0.12
Seizure type							
Focal onset	16 (57.14)	11 (39.29)	1 (3.57)	15 (53.57)	12 (42.86)	0	1 (3.57)
Generalized onset	16 (48.48)	14 (42.42)	3 (9.09)	16 (48.48)	9 (27.27)	3 (9.09)	5 (15.15)
Unknown	7 (38.89)	7 (38.89)	4 (22.22)	8 (44.44)	2 (11.11)	3 (16.67)	5 (27.78)
*p*-value			0.38				0.032
Embolization							
Partial	49 (50)	39 (39.8)	10 (10.2)	49 (50)	30 (30.61)	5 (5.1)	14 (14.29)
Total	11 (61.11)	7 (38.89)	0	9 (50)	6 (33.33)	2 (11.11)	1 (5.56)
*p*-value			0.44				0.59
Clinical manifestation							
With hemorrhage	19 (54.29)	13 (37.14)	3 (8.57)	17 (48.57)	12 (34.29)	1 (2.86)	5 (14.29)
Seizure only	41 (50.62)	33 (40.74)	7 (8.64)	41 (50.62)	24 (29.63)	6 (7.41)	10 (12.35)
*p*-value			0.95				0.82
Years with seizures							
<5	25 (86.21)	4 (13.79)	0	24 (82.76)	5 (17.24)	0	0
5–10	29 (49.15)	28 (47.46)	2 (3.39)	28 (47.46)	25 (42.37)	4 (6.78)	2 (3.39)
>10	5 (18.52)	14 (51.85)	8 (29.63)	5 (18.52)	6 (22.22)	3 (11.11)	13 (48.15)
*p*-value			<0.001				<0.001

## Data Availability

All the data collected and used for this study are available for public viewing via the link https://zenodo.org/record/7329781, accessed on 17 November 2022.

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
