# Peer review of "Effect of Targeted Embolization on Seizure Outcomes in Patients with Brain Arteriovenous Malformations"

_diagnostics, 2022, doi:10.3390/diagnostics13010047_

Round 1

Reviewer 1 Report

In this study, the authors investigate the the outcome of curative embolization on patients with brain arteriovenous malformations with seizures. Overall, this study is interesting. Some issues should be addressed which are listed below:

Major:

(1)   Please furture emphasize the motivation and contribution of this study to the field. What is the clinical significance of this study?

(2)   In the Figure of patient exclusion and inclusion flow chart, 179 patients with complete follow-up and 54 patients did not have seizures, why 137 eligible patients remained? 179-54 = 137? Again, 137 patients and 9 died, but 116 patients were included. 137-9 = 116?

(3)   Clinical outcomes were tested for association with independent variables with one-way analysis of variance (ANOVA) and Fisher’s exact test. Generally, ANOVA compares the means of two or more independent groups. Therefore, the authors should clarify they use these statistical analyses to compare what? Which groups?

Minor:

(1)   A space is needed before reference, please check thoughout the manuscript. For example, …when compared to multimodal treatment or conservative treatment[9]. …78% of postoperative seizure freedom[12].

(2)   A total of 421 patients, who underwent embolization were reviewed for eligibility to be included in the study (Figure). Figure should be Figure 1?

(3)   Lack of punctuation. For example, Only 15.5% of patients have achieved complete embolization of the 150 AVM, the rest of the patients had only partial embolization

(4)   Fischer’s exact test should be Fisher's exact test

Author Response

Dear Reviewer,

We want to thank you for your time and valuable comments. Your suggestions have been duly noted and addressed to the best of our abilities. The responses to the comments are listed below:

Major:

(1) Please further emphasize the motivation and contribution of this study to the field. What is the clinical significance of this study?

Response: The clinical significance of this study is that with careful consideration endovascular embolization can be an effective treatment method for brain AVMs presented with short-term nature seizures. The importance of the study was added in lines 72-73 and 319-320.

(2)   In the Figure of patient exclusion and inclusion flow chart, 179 patients with complete follow-up and 54 patients did not have seizures, why 137 eligible patients remained? 179-54 = 137? Again, 137 patients and 9 died, but 116 patients were included. 137-9 = 116?

Response: unfortunately, a mistake was made while filling out the flow chart, 179-54 = 125, 125-9 = 116. The flow chart was edited, and the correct version was included

(3)   Clinical outcomes were tested for association with independent variables with one-way analysis of variance (ANOVA) and Fisher’s exact test. Generally, ANOVA compares the means of two or more independent groups. Therefore, the authors should clarify they use these statistical analyses to compare what? Which groups?

Response: the ANOVA test was used to compare mean age of patients with positive, moderate, and negative ILAE score between each other, as well as age of patients with Engel class 1, 2, 3, and 4. All the groups were independent of each other.

 Minor:

(1) A space is needed before reference, please check thoughout the manuscript. For example, …when compared to multimodal treatment or conservative treatment[9]. …78% of postoperative seizure freedom[12].

Response:  Spaces before references were inserted throughout the manuscript.

(2) A total of 421 patients, who underwent embolization were reviewed for eligibility to be included in the study (Figure). Figure should be Figure 1?

Response: a mistake has been fixed

(3) Lack of punctuation. For example, Only 15.5% of patients have achieved complete embolization of the 150 AVM, the rest of the patients had only partial embolization

 Response: a mistake has been fixed

(4) Fischer’s exact test should be Fisher's exact test

Response: a mistake has been fixed

Reviewer 2 Report

In this paper, the Author present their experience with AVM patients treated endovascularly and concentrate their focus on the seizure outcome. In their analysis, the only factor significantly associated with post-treatment improvement was a medical history of a relatively short period with epilepsy. 

I first of all would like to congratulate the Authors with the manuscript they provided: despite some (relevant) English language editing being necessary, their work has well-stated purposes, the methods and the results are clearly and concisely presented, the discussion touches the most relevant points on this issue, the limitations are honestly and clearly stated. No redundant data or analysis are present. It is opinion of this Reviewer that any scientific work should display all these features, as this work does.

Concerning the content of the paper, I think that it deserves consideration for publication. In fact, despite the limitations of the study and the results which could be explained by pathological white matter networks resulting from long-lasting epilepsy which persist despite AVM treatment, it is interesting and thought-provoking to propose that early endovascular treatment could be helpful in improving seizure outcome.

To further improve the interest of the paper, I would like to suggest a small improvement. The Authors analyzed SM grades, but, as we know, this grading is the result of three items, of which AVM size and venous drainage patterns could actually play a role in seizure outcome. Have the Authors thought about analyzing this two features independently as potential predictive factors?

Minor things:

- I would suggest to improve Table 4 editing for better readabilty

- English editing is warmly suggested

Author Response

We want to thank the Reviewer for their valuable comments and recommendations, which helped us to improve our manuscript. The response to the comments are provided below:

We understand that superficial cortical drainage is associated with seizures, however, in our study, we had a mix of superficial and deep drainage cases. Unfortunately, in this study, the AVM size and venous drainage patterns were not analyzed separately. The characteristics were considered when assigning the AVM grade but were not recorded separately in the database, therefore, the analysis of the two features independently is currently not possible. We will consider mentioned suggestion for further studies to avoid such shortcomings.

Edits to Table 4 stylization were made for better readability.

English editing has been performed to the best of our abilities.

We want to thank the Reviewer again for their time and valuable suggestions!

Round 2

Reviewer 1 Report

Accept as is.